# Information Directed Tree Search
## Reasoning and Planning with Language Agents

**Yash Chandak, Hyunji Alex Nam, Allen Nie, Jonathan Lee and Emma Brunskill**
Department of Computer Science
Stanford
Stanford, CA
`<ychandak@stanford.edu>`

## Abstract

Solving challenging tasks often require agentic formulation of language models that can do multi-step reasoning and progressively solve the task by collecting various feedback. For computational efficiency, it may be advantageous to quantify the information associated with different feedback and guide the search such that the solution can be obtained quickly. To explore this possibility, we take a Bayesian approach and propose an *information directed tree search* (IDTS) algorithm that makes use of in-context learning to approximate the information associated with different feedback. We explore the effectivity of IDTS on challenging tasks involving programming, formal math, and natural language. Interestingly, while we find advantages over simple uniform search methods, the proposed approach is about comparable to MCTS even though it explores different paths. We discuss some possibilities for our findings and highlight open questions for future work.

## 1 Introduction

Large language models (LLMs) have become integral for building autonomous agents that aim to find solutions to challenging tasks. Many such tasks often require multi-step reasoning, are not well specified, or require hierarchical decomposition [Hao et al., 2023, Zelikman et al., 2022]. In such cases, even though LLMs cannot directly provide the complete answer, they can provide partially correct responses, or answer sub-parts of the prompt. Such responses can be then paired with other source(s) of feedback that can subsequently guide the reasoning of the language models. For instance, if the generated response is a code, then the feedback can be provided by program compilers and unit testing [Zhang et al., 2023a, Zhou et al., 2023]. If the generated response is formal proofs for mathematical statements, then auto-verifiers can be used [First et al., 2023, Zheng et al., 2023]. Further, language models can also be used to self-critique their responses [Zhou et al., 2023].

However, such sources of feedback vary in terms of their quality. For instance, compilers can only flag incorrect code - they cannot tell what to change in that incorrect code. In contrast, while language model critics can suggest what to change, they might often hallucinate and provide inaccurate feedback. Similarly, humans may want to work with a language model to achieve a goal but may not know themselves what the right steps to reach that goal are. This raises the main question of interest: How do we leverage partially correct response generating language models, with partially correct source(s) of feedback, to find solution to important problems?

We cast this as a planning problem with partially correct feedback, and design a new tree search procedure for inference time planning. To be computationally efficient when resources are limited, unlike MCTS [Coquelin and Munos, 2007, Kocsis and Szepesvári, 2006] that does a naive count-

Workshop on Bayesian Decision-making and Uncertainty, 38th Conference on Neural Information Processing Systems (NeurIPS 2024).

based exploration, we take a Bayesian approach and prioritize search towards feedback that provides higher *information gain* towards the solution.

| | LLMs | Planning | Robust | Rich Exploration |
|---|:---:|:---:|:---:|:---:|
| One-shot generation | ✓ | ✗ | ✗ | ✗ |
| Iterative Refinement/CoT | ✓ | ✓ | ✗ | ✗ |
| Search (MCTS, best-first, etc.) | ✓ | ✓ | ✓ | ✗ |
| IDTS (Proposed) | ✓ | ✓ | ✓ | ✓ |

Table 1: Methods that have single chain of thought are often not robust when the feedback is only partially correct. In contrast, search based methods maintain several solutions and corresponding feedback, and are thus less likely to be derailed in their reasoning if the feedback is not perfect. However, conventional search based methods still explore using count-based statistics, which fail to account for any notion of information associated with each feedback. The proposed IDTS method aims at alleviating all these issues. We discuss related work in more detail in Appendix A.

**Problem Statement:** Let $o \in \mathcal{O}$ and $a \in \mathcal{A}$ correspond to an observation and an action. let each observation consist of some context $x \in \mathcal{X}$ and a reward $r \in \mathbb{R}$, such that $o = (x, r)$. Let $O_i = (X_i, R_i)$ and $A_i$ be the random variables corresponding to observation and action at interaction step $i$. Specifically, the context $X_0$ in the initial observation $O_0$ contains the question, and the action $A_i$ is an agent's attempt at the solution and $X_{i+1}$ contains rich feedback on that action. The goal of the agent is to determine an answer $Y \in \mathcal{Y}$ to that question within a fixed number of interactions. For example, for coding tasks, $X_0$ is a coding question, $A_i$ is the language model's code, $Y$ is a correct code solution, and $O_{i+1}$ contains the error messages from compiler and results from unit tests for $A_i$.

We define a state $s \in \mathcal{S}$ to be the history of the interaction so far, i.e., $S_i := (O_0, A_0, O_1, A_1, \ldots, O_i)$. Let $\pi : \mathcal{S} \to \mathcal{T}(\mathcal{A})$ be a policy and let $\tau^\pi(s)$ correspond to the trajectory unrolled using policy $\pi$ starting from a state $s$. In this work, we build upon pretrained LLMs and thus implicitly leverage side information (e.g., data on the internet) to find the solution $Y$. Let $\mathcal{D}$ denote such data. Let the random variable $\mathcal{T}_t$ be the tree containing everything observed till the end of the $t^{\text{th}}$ iteration. With a slight abuse of notation, we will use a subscript of $t$ to denote implicit conditioning on everything observed so far, i.e., $\mathcal{T}_t$ and $D$. For example, $\mathbb{P}_t(Y = y|x) := \mathbb{P}(Y = y|x, \mathcal{T}_t, D)$.

**Information Directed Sampling:** Given random variables $X$ and $Y$, the information gain for $Y$ on observing $X$ is $\mathcal{I}_t(Y; X) := \mathcal{H}_t(Y) - \mathcal{H}_t(Y|X)$, where $\mathcal{H}_t(Y) := -\sum_{y \in \mathcal{Y}} \mathbb{P}_t(y) \log \mathbb{P}_t(y)$ is the entropy and $\mathcal{H}_t(Y|X) := -\sum_{x \in \mathcal{X}} \mathbb{P}_t(x) \sum_{y \in \mathcal{Y}} \mathbb{P}_t(y|x) \log \mathbb{P}_t(y|x)$ is the conditional entropy.

In the bandit setting, let $Z(A) := (O, A)$ be the rich observation (potentially including reward), and action $A = a$ on executing $a$. For the optimal action $a^*$, let $\mathcal{I}_t(a^*; Z(a))$ corresponds to the information gain towards the agent's belief over the optimal action $a^*$ given $Z(a)$. Similar to Bayesian experimental design [Rainforth et al., 2024], information directed sampling[Russo and Van Roy, 2014] or its $\lambda$-regularized version [Hao and Lattimore, 2022] selects an action such that it maximizes the performance as well the information gained by the agent,

$$a \in \arg\max_{a \in \mathcal{A}} \frac{\mathcal{I}_t(a^*; Z(a))}{\mathbb{E}_t[r(a^*) - r(a)]^2}, \qquad a \in \arg\max_{a \in \mathcal{A}} \ r(a) + \lambda \mathcal{I}_t(a^*; Z(a)). \qquad (1)$$

**Monte-Carlo Tree Search:** A popular planning procedure is MCTS that tracks $N_t(s, a)$ and $N_t(s)$, i.e., the number of times a particular $(s, a)$ pair and $(s)$ has been chosen during tree traversal till iterate $t$. Let $\mathcal{T}_t$ be the tree expanded at the start of a given iterate $t$. At every iterate $t$, one of the nodes of $\mathcal{T}_t$ is chosen to be expanded by selecting action $a \in \mathcal{A}$ for a given state $s \in \mathcal{S}$. Most MCTS algorithms are based on the seminal upper confidence tree (UCT) algorithm [Kocsis and Szepesvári, 2006, Coquelin and Munos, 2007] that chooses

$$a \in \arg\max_{a \in \mathcal{A}} \left\{ \hat{q}_t(s, a) + c\sqrt{\frac{\log N_t(s)}{N_t(s, a)}} \right\}, \qquad (2)$$

where $\hat{q}$ is the estimate of the cumulative return [Sutton and Barto, 2018], and the term in blue guides the exploration. While (2) provides one approach to balance exploration and exploitation based on counts, not only such discrete notion of counts fail to account for similarities between different states, it also fails to incorporate richer notion of explorations, e.g., even if an agent has taken an action

$a \in \mathcal{A}$ often, it might still want to explore that action more if the *information* being obtained from doing so is large.

The key insight of this work is to leverage ideas from (1) and use information gain to improve the exploration technique for monte-carlo tree search (2). Particularly, for our problem setup, we let the action $a$ be a complete sentence/solution generated by a LLM, and $s$ be the history of past interactions. Then, if the task is related to coding, we want to prioritize a solution $a$, which together with its error messages and unit-tests $Z(a)$ provides higher information gain towards the solution $a^*$. (While in principle, maximization over the action set $\mathcal{A}$ in intractable, similar to prior work [Zhou et al., 2023, Jang et al., 2020] we let $\mathcal{A}$ be k sampled completions for any given state $s$.) In the remaining sections, we discuss how to estimate information gain associated with taking any action $a$ in state $s$.

## 2 Information Directed Tree Search

Central to our idea is a procedure to estimate information gain (IG), such that the proposed idea can scale and be used with LLMs. Here we focus on explaining the core idea of computing IG using in-context learning for a single interaction step. Due to space constraints, in Appendix B, we discuss how to use chain-rule of IG to recursively decompose IG over a *sequence of actions* under a policy.

Recall that a state $s$ is the history of interactions, and let $s' = (s, a, o)$ be the subsequent state on enacting $a$. The information gain of taking an action $a$ and observing the new state $s'$ is the reduction in the uncertainty of the agent's belief over the solution $Y$, i.e., $\mathcal{I}_t(Y; s') = \mathcal{H}_t(Y) - \mathcal{H}_t(Y|s')$.

Computing $\mathcal{I}_t(Y; s')$ requires estimating entropies $\mathcal{H}_t(Y)$ and $\mathcal{H}_t(Y|s')$, which in turn depend on the probability distributions $\mathbb{P}_t(y)$ and $\mathbb{P}_t(y|s')$ for $\forall y \in \mathcal{Y}$, respectively. In conventional RL methods, this requires updating the posterior given the new state $s'$. Specifically, recall $\mathbb{P}_t(y|s') = p(Y = y|s', \mathcal{T}_t, D)$, where $\mathcal{T}_t$ is the tree explored till the iterate $t$, and let $\mathcal{T}_{t+1} = (\mathcal{T}_t \cup s')$ be the tree for the next iterate after observing $s'$. Let $\phi \in \Phi$ be the parameters of the environment model for the problem the agent is facing. Here, the posterior updates entail computing

$$p(y|s', \mathcal{T}_t, D) = p(y|\mathcal{T}_{t+1}, D) = \int p(y|x_0, \phi)p(\phi|\mathcal{T}_{t+1}, D)\mathrm{d}\phi. \tag{3}$$

Unfortunately, such posterior updates can be challenging when $\phi$ is high-dimensional. The challenge is particularly exacerbated because posterior needs to be *repeatedly* updated for $\mathcal{T}_{t+k}$, where $k \geq 1$, as new data is acquired, thereby making prior methods intractable beyond simple/linear settings.

**In-context learning for posterior updates:** To mitigate this challenge, we build upon the recent insights [Xie et al., 2021, Lee et al., 2023] that draw connections between in-context learning and Bayesian inference. In our setting, in-context learning provides a remarkably simple posterior update for $p(y|\mathcal{T}_{t+1}, D)$. Specifically, consider the model in Figure 1. Let $\theta$ be the parameters of the general world model, and $\phi$ be the model parameters for the specific problem agent is dealing with, and $D$ is the internet data. We will make the following assumption,

**Assumption 1.** $\forall t, \mathcal{I}(\theta; \mathcal{T}_t|D) = 0.$

Figure 1: Graphical model for the data generating process.

Assumption 1 states that given the internet-scale data $D$, a few (since $t$ is usually small) problem specific interactions contained in $\mathcal{T}_t$ do not provide any more information about the general world model $\theta$. This implies $p(\theta|\mathcal{T}_t, D) = p(\theta|D)$. Under this viewpoint we can express $p(y|\mathcal{T}_{t+1}, D)$ as the following,

$$p(y|\mathcal{T}_{t+1}, D) = \iint p(y, \phi, \theta|\mathcal{T}_{t+1}, D)\mathrm{d}\theta\mathrm{d}\phi = \int \left( \int p(y, \phi|\mathcal{T}_{t+1}, \theta, D)\mathrm{d}\phi \right) p(\theta|\mathcal{T}_{t+1}, D)\mathrm{d}\theta \tag{4}$$

$$= \int p(y|\mathcal{T}_{t+1}, \theta, D)p(\theta|\mathcal{T}_{t+1}, D)\mathrm{d}\theta \stackrel{(a)}{=} \int p(y|\mathcal{T}_{t+1}, \theta)p(\theta|D)\mathrm{d}\theta, \tag{5}$$

where (a) follows because of the model in Figure 1, and Assumption 1. Similarly, $\mathbb{P}_t(y) = p(y|\mathcal{T}_t, D)$ needed to compute $\mathcal{H}_t(Y)$ can be factorized as $p(y|\mathcal{T}_t, D) = \int p(y|\mathcal{T}_t, \theta)p(\theta|D)\mathrm{d}\theta$. In Appendix B we discuss how once the posterior $\mathbb{P}_t(y|s')$ and $\mathbb{P}_t(y)$ are available, we can readily estimate the entropies $\mathcal{H}_t(Y|s')$ and $\mathcal{H}_t(Y)$, and thus also the information gain $\mathcal{I}_t(Y; s')$.

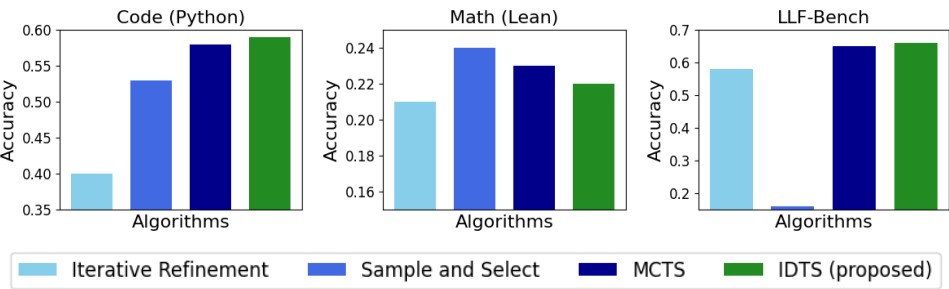

Figure 2: We compare the following baselines across domains: **MCTS:** standard MCTS with UCT style bonus, **Iterative Refinement:** This is sequential interaction (equivalent to MCTS with branching factor=1), **Sample and select:** This samples k solutions at the root node, and effectively disregards any feedback (equivalent to MCTS with tree depth=1), **IDTS:** This is the proposed algorithm, that builds on MCTS but uses information gain to drive exploration as opposed to the UCT style bonus. The maximum number of nodes expanded is 10. For MCTS and IDTS, the branching factor is 4.

**Advantages:** Unlike $p(\phi|\mathcal{T}_{t+1}, D)$ in (3) that does an explicit update to obtain posterior over $\phi$ for the underlying environment, in (5) the posterior is over the parameters of the world-model $p(\theta|D)$ and thus do *not* require an *explicit* update when new data is acquired. Instead, the new data is used *in-context* $p(y|\mathcal{T}_{t+1}, \theta)$ to obtain the desired $p(y|\mathcal{T}_{t+1}, D)$. Not only this avoid any updates to the parameters $\theta$, but also as new data becomes available then $p(y|\mathcal{T}_{t+k}, D)$ can also be computed readily for $k \geq 1$. *This would not have been possible without the in-context learning ability of LLMs.*

Further, (5) reduces the problem to standard uncertainty estimation for deep learning [Gawlikowski et al., 2021, Abdar et al., 2021]. Perhaps the most popular technique is to use an ensemble of N models for a Monte-Carlo estimate of the integral in (5). For extremely large models, ensembles can be created using multiple-low rank adapter instead [Malinin and Gales, 2020, Kuhn et al., 2023]. For simplicity, we use $N = 1$ in our experiments.

## 3  Empirical Analysis

Complete algorithm for the proposed IDTS method and more details about the experimental setup is provided in Appendix C. We run all the experiments using OpenAI GPT models [Achiam et al., 2023], and the results are presented in Figure 2. We compare the performances of the above methods on the following domains:

**Code (Python):** This is based on the HumanEval benchmark [Chen et al., 2021]. We focus on the hard problems by filtering out the ones that can be solved by GPT-3.5-turbo in one-shot. Here, an action corresponds to the entire solution. The feedback consists of results from the synthetically generated (and thus potentially incorrect) unit-tests, error messages from the compiler, and self-critique of the solution given the unit-test results and error messages.

**Math (Lean):** We use the MiniF2F benchmark [Zheng et al., 2021] for this task. Each action corresponds to an entire formal proof, and the feedback consists of the error messages from the Lean [Moura and Ullrich, 2021] compiler along with the self-critique of the generated solution.

**LLF-Bench:** [Cheng et al., 2023] This task requires recommending movies to a user whose interests are hidden from the agent. After every suggestion, the domain provides natural language hint regarding how close the recommended movies are to the user's interests. In the feedback, we also incorporate self-critique of the solution based on the response received.

While IDTS shows some promise on coding and LLF-bench, the gains over MCTS are not significant at the moment to justify additional complexity associated with IDTS. Math domain stands out, as neither the error messages from Lean compiler had corrective feedback, nor the self-critique provided any meaningful feedback. As such, just ignoring any feedback and sampling diverse solutions emerged to be the best strategy in this domain.

# 4  Discussion and Future Work

While in principle, having access to the true IG should increase the efficiency of the search significantly, it is not feasible at the LLM scale to obtain the true IG. IDTS avoids (repeated) posterior updates using in-context learning, but still requires useful uncertainty measure for LLMs [Malinin and Gales, 2020, Quach et al., 2023]. An important direction for future work is to have a deeper study on the estimation error in the IG computation, as finding other better methods of IG estimation is required before using IG is likely to be helpful with LLMs.

Note, however, that search using IG only requires deciding which node to explore. As such, it might only be needed that the *relative* (not absolute) values of the IG across nodes are accurate. While this could potentially mitigate the IG estimation challenge, assessing the quality of ranking also remains challenging as the true rankings are unknown. Further, expanding $\gg 10$ nodes per tree would provide invaluable insights on the utility of different exploration strategies.

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

# Information Directed Tree Search
## (Appendix)

## A Related Work

### A.1 LLM + MCTS based Reasoning and Planning

Chain of thought reasoning [Wei et al., 2022] and self improvement [Zelikman et al., 2023] have shown a lot of success in the past. Building on the success of these tree of thought [Yao et al., 2023] maintains a diverse set of solutions. Alternatively, planning via tree search based algorithms have also been studied [Hao et al., 2023, Zhou et al., 2023]. Extensions of $A^*$ search [Zhuang et al., 2023], or even in-context tree traversal [Zhang et al., 2023b], and hierarchical search [Parascandolo et al., 2020, Zelikman et al., 2022, Wang et al., 2023b] have also been explored.

Several methods have considered solving formal math problems using different variants of MCTS and best-first search [Yang et al., 2023, Limperg and From, 2023, Wang et al., 2023a, Lample et al., 2022, Zhu et al., 2022], doing proof repair using feedabck (akin to iterative-refinement) [Zheng et al., 2023, First et al., 2023], by hierarchical decomposition [Xin et al., 2023], or incremental learning [Aygün et al., 2022, Polu et al., 2022]. MCTS for language based RL games [Jang et al., 2020] and code-generation [Zhang et al., 2023a] has also been considered.

The approaches developed in these are largely complementary to ours, but none of them focus on the topic of how to explore more efficiently in the presence of partially correct feedback.

### A.2 Information Directed Sampling

Building upon the initial work on IDS [Russo and Van Roy, 2014], the core idea has been further expounded by several [Lu et al., 2023, Hao et al., 2022, Hao and Lattimore, 2022] with a focus on theoretical aspects of the regret associated with various instances of the IDS idea. On the practical side, Zanette and Sarkar [2017] considered the tabular setting and aimed at a tractable approximation of the information ratio using a variance based bound for IG. From one perspective, if we could have replaced the information gain term in the proposed IDTS method with a variance based bound, then our method would come closer to using Bernstein based bonuses for tree search [Lieck et al., 2017].

Kirschner and Krause [2018] discuss how in Gaussian setting, IG based on Bayesian posterior distribution can be expressed in frequentist terms. Nikolov et al. [2018] built upon this to consider the generic/deepRL setting and estimate both the aleatoric (using distributional RL) and epistemic uncertainty (using a q-ensemble). IDS has also been used for doing open/closed loop planning by changing the 'reward' function to consider the information gained about the optimal trajectory by running the desired sequence of actions Mehta et al. [2021, 2022]. While related, none of them tackle the LLM setting.

Our work is also related to works that discuss what information to gather [Arumugam and Van Roy, 2021a], and how the acquisition of it can be made faster [Arumugam and Van Roy, 2021b]. Similarly, it also related to the work on reformulating POMDPs for optimizing long term value of information when eliciting preferences [Boutilier, 2002] and query optimization [Chajewska et al., 2000, Boutilier, 2013, Biyik et al., 2023]. Several recent works also consider the task of strategically collecting labels for which pair of outputs to query in order to minimize the number of samples need for RLHF training [Mehta et al., 2023, Freedman et al., 2023, Ji et al., 2024, Das et al., 2024] or for in-context learning [Margatina et al., 2023]. Our work complements this direction and considers strategic interaction at the inference time to find a solution to given question, and involves no RLHF.

## B Information Gain Estimation

### B.1 Preliminaries

Let $X$ and $Y$ be two random variables. Recall, from the chain rule of entropy:

$$\mathcal{H}(X, Y) = \mathcal{H}(X) + \mathcal{H}(Y|X) \qquad \text{(Chain rule of entropy)} \qquad (6)$$

Similarly, information gain between random variables $\{X_1, X_2\}$ and $Y$ can be written as,

$$\mathcal{I}(Y; X_1, X_2) = \mathcal{H}(Y) - \mathcal{H}(Y|X_1, X_2) \tag{7}$$
$$= \mathcal{H}(Y) - \mathcal{H}(Y|X_1) + \mathcal{H}(Y|X_1) - \mathcal{H}(Y|X_1, X_2) \tag{8}$$
$$= \mathcal{I}(Y; X_1) + \mathcal{I}(Y; X_2|X_1). \quad \text{(Chain rule of information gain)} \tag{9}$$

## B.2 Intra-interaction decomposition:

Recall that $p(y|\mathcal{T}_{t+1}, D)$ from (5) was required to compute the information gain $\mathcal{I}_t(Y; s')$,

$$p(y|\mathcal{T}_{t+1}, D) = \int p(y|\mathcal{T}_{t+1}, \theta) p(\theta|D) \mathrm{d}\theta, \tag{10}$$

Here, the random variable $Y$ that characterizes the agent's belief over the solution is a sequence of token $(Y_0, Y_1, \ldots)$ that is generated auto-regressively. This structure can be leveraged to specify an estimator that decomposes the information gain at per token level,

$$\hat{\mathcal{I}}_t(Y; s') := -\sum_{i=1}^{K_2} \log \mathbb{P}_t(Y_i|Y_{:i-1}) + \sum_{i=1}^{K_1} \log \mathbb{P}_t(Y_i|Y_{:i-1}, s'), \tag{11}$$

where $\mathbb{P}_t(Y_i|Y_{Y_{:i-1}})$ is a random variable because conditioning is on a random variable as well. Further, notice that the two sequence of $Y_i$'s are different, as one is conditioned on $s'$ and the other is not. $K_1$ and $K_2$ are the lengths of the generated sentences.

Estimator $\hat{\mathcal{I}}_t(Y; s')$ decomposes information gain such that its computation requires logprobs of the generated tokens, which is accessibly even from proprietary large-language models like GPT [OpenAI, 2023]. However, when using open-source models, notice that the agent has additional access to the logprobs of the tokens *not* sampled. These can be leveraged to create the following improved estimator that can have lower variance than $\hat{\mathcal{I}}_t(Y; s')$, but still provides an unbiased estimator of $\mathcal{I}_t(Y; s')$.

$$\widetilde{\mathcal{I}}_t(Y; s') := \sum_{i=1}^{K_2} \mathcal{H}_t(Y_i|Y_{:i-1}) - \sum_{i=1}^{K_1} \mathcal{H}_t(Y_i|Y_{:i-1}, s'). \tag{12}$$

It can be observed using chain-rule of entropy and tower law of expectations that,

$$\mathbb{E}\left[\widetilde{\mathcal{I}}_t(Y; s')\right] = \mathbb{E}\left[\hat{\mathcal{I}}_t(Y; s')\right] = \mathcal{I}_t(Y; s'). \tag{13}$$

## B.3 Inter-interaction decomposition:

let $\pi$ be a tree-traversal policy that operates on the tree $\mathcal{T}_t$ (This policy has a composite nature, where it either selects one of the already expanded branches, or draws a new sample to create a new branch in the tree). For any given node $s$, let $\tau^\pi(s)$ be a random trajectory that could be unrolled using $\pi$ when starting at node $s$. Let $\mathcal{L}_t(s)$ be the set of leaves under the *subtree* of $s$ in $\mathcal{T}_t$. Let $\mathcal{C}_t(s)$ be the set of immediate children of $s$, and $\mathcal{A}_t(s)$ be all the ancestors of $s$ in $\mathcal{T}_t$.

In the following Theorem 1, we formalize a procedure to recursively estimate the information gain on expanding the sub-tree of a given node $s$. This has an intuitive form that asserts that the information gain from expanding the subtree of a node $s$ equals the expected information gain from expanding the children of node $s$.

**Theorem 1.** *The information gain from $\tau_t(s)$ can be expressed as the following recursive form,*

$$\mathcal{I}_t(Y; \tau^\pi(s)) = \sum_{s' \in \mathcal{C}_t(s)} \mathbb{P}_t(s'|s; \pi) \mathcal{I}_t(Y; \tau^\pi(s')) = \sum_{x \in \mathcal{L}_t(s)} \mathbb{P}_t(x|s; \pi) \mathcal{I}_t(Y; \tau^\pi(x)). \tag{14}$$

*Proof.* Consider the following decomposition $\tau(s) = \{s \cup \tau_2\}$, where $\tau_2$ is the part of the trajectory $\tau(s)$ without node $s$. Note that the first node in $\tau_2$ will be a $s' \in \mathcal{C}_t(s)$.

$$\mathcal{I}_t(Y; \tau(s)) = \mathcal{H}_t(Y) - \mathcal{H}_t(Y|\tau(s)) \tag{15}$$

Considering the term in red in 15,

$$\mathcal{H}_t(Y|\tau(s)) = \mathcal{H}_t(Y|s, \tau_2) \tag{16}$$

$$= \sum_{x,\tau} \mathbb{P}_t(s = x)\,\mathbb{P}_t(\tau_2 = \tau|s = x)\,\mathcal{H}_t(Y|x,\tau) \tag{17}$$

$$\overset{(a)}{=} \sum_{\tau} \mathbb{P}_t(\tau_2 = \tau|s)\,\mathcal{H}_t(Y|s,\tau) \tag{18}$$

$$= \sum_{s' \in \mathcal{C}(s)} \mathbb{P}_t(s'|s) \sum_{\tau} \mathbb{P}_t(\tau_2 = \tau|s,s')\,\mathcal{H}_t(Y|s,\tau) \tag{19}$$

$$\overset{(b)}{=} \sum_{s' \in \mathcal{C}(s)} \mathbb{P}_t(s'|s) \sum_{\tau} \mathbb{P}_t(\tau(s') = \tau)\,\mathcal{H}_t(Y|s,\tau) \tag{20}$$

$$= \sum_{s' \in \mathcal{C}(s)} \mathbb{P}_t(s'|s)\mathcal{H}_t(Y|s,\tau(s')) \tag{21}$$

where $(a)$ follows as $s$ is the first node in the trajectory $\tau(s)$ deterministically. $(b)$ follows as conditioned on $s'$, $\tau_2$ does not depend on $s$. Now, focusing on the blue term in 21,

$$\mathcal{H}_t(Y|s, \tau(s')) = -\sum_{x,\tau} \mathbb{P}_t(s = x)\,\mathbb{P}_t(\tau(s') = \tau|s = x) \sum_y \mathbb{P}_t(y|x,\tau) \log \mathbb{P}_t(y|x,\tau) \tag{22}$$

$$\overset{(a)}{=} -\sum_{\tau} \mathbb{P}_t(\tau(s') = \tau) \sum_y \mathbb{P}_t(y|s,\tau) \log \mathbb{P}_t(y|s,\tau) \tag{23}$$

$$= -\sum_{\tau} \mathbb{P}_t(\tau(s') = \tau) \sum_y \mathbb{P}(y|s,\tau,\mathcal{T}_t) \log \mathbb{P}(y|s,\tau,\mathcal{T}_t) \tag{24}$$

$$\overset{(b)}{=} -\sum_{\tau} \mathbb{P}_t(\tau(s') = \tau) \sum_y \mathbb{P}(y|\tau,\mathcal{T}_t) \log \mathbb{P}(y|\tau,\mathcal{T}_t) \tag{25}$$

$$= -\sum_{\tau} \mathbb{P}_t(\tau(s') = \tau) \sum_y \mathbb{P}_t(y|\tau) \log \mathbb{P}_t(y|\tau) \tag{26}$$

$$= -\sum_{\tau} \mathbb{P}_t(\tau(s') = \tau)\,\mathcal{H}_t(Y|\tau) \tag{27}$$

$$= \mathcal{H}_t(Y|\tau(s')) \tag{28}$$

where $(a)$ follows because $s$ and $s'$ are fixed variables in the LHS, and the trajectory $\tau(s')$ does not depend on ancestor node $s$ of $s'$. $(b)$ follows because by construct ancestor $s$ is a part of $\mathcal{T}_t$, i.e., $s \in \mathcal{T}_t$ already. Therefore, combining 15, 21, and 28,

$$\mathcal{I}_t(Y; \tau(s)) = \mathcal{H}_t(Y) - \sum_{s' \in \mathcal{C}_t(s)} \mathbb{P}_t(s'|s)\,\mathcal{H}_t(Y|\tau(s'))$$

$$= \sum_{s' \in \mathcal{C}_t(s)} \mathbb{P}_t(s'|s)[\mathcal{H}_t(Y) - \mathcal{H}_t(Y|\tau(s'))]$$

$$= \sum_{s' \in \mathcal{C}_t(s)} \mathbb{P}_t(s'|s)\,\mathcal{I}_t(Y; \tau_t(s')).$$

Now, unrolling $\mathcal{I}_t(Y; \tau_t(s'))$ using the above recursion and observing that for a leaf node $x \in \mathcal{L}_t(s)$, $\tau(x) = x$, we obtain the aforementioned result.

$\square$

Contrast this recursive propagation of information gain, with recursive propagation of the (terminal) reward. Information gain from intermediate states in a trajectory is zero, similar to intermediate rewards for many games. This permits minimal modification of existing MCTS methods to incorporate rich feedback and perform an information-directed exploration.

## B.4 Information Gain Bellman Recursion

Let $\mathscr{I}_t^\pi(s)$ be the information gained towards the optimal solution $y$ on unrolling a trajectory $\tau^\pi(s)$, given the agent is at state $s$ during iteration $t$. Also, recall that the sub-script of $t$ denotes implicit conditioning on all the data observed so far (e.g., in the code-gen setting, it denotes conditioning on all the internet data that has been used for model training),

$$\mathscr{I}_t^\pi(s) := \mathcal{I}_t(Y; \tau^\pi(s)|s). \tag{29}$$

One should consider $\mathscr{I}_t^\pi(h)$ analogous to the state-value function for a policy $\pi$, but instead of being for the long-term return to go, it is for the long-term information gain. Similar to the state-value function, it is also possible to define a Bellman recursion for the information gain (recall that by construct, a state is defined using the history of the trajectory).

**Theorem 2.** *Information gain recursion:*

$$\mathscr{I}_t^\pi(s) = \mathcal{I}_t(Y; S'|s) + \mathbb{E}_\pi[\mathscr{I}_t^\pi(S')|s], \tag{30}$$

*where $S'$ is observed following the policy $\pi$ from state $s$.*

**Corollary 1.** *n-step Information gain backup*

$$\mathscr{I}_t^\pi(s) = \mathbb{E}_\pi\left[\sum_{i=0}^{n-1} \mathcal{I}_t(Y; S_{i+1}|S_i) + \mathscr{I}_t^\pi(S_n)|s\right], \tag{31}$$

*where $S_0 = s$.*

*Proof.* Follows by unrolling the recursion in (30). $\qquad\square$

*Proof.* Proof of Theorem 2. Without loss of generality, consider $s = (O_0)$, i.e., just the starting state, and consider the length of a trajectory to be $L$,

$$\mathscr{I}_t^\pi(s) = \mathcal{I}_t(Y; \tau^\pi(s)|s) \tag{32}$$

$$= \mathcal{I}_t(Y; (A_0, O_1, A_1, \ldots, O_L)|s) \tag{33}$$

$$= \mathcal{I}_t(Y; S'|s) + \mathcal{I}_t(Y; (A_1, \ldots, O_L)|s, S') \qquad \because \text{Chain rule of info gain} \tag{34}$$

$$= \mathcal{I}_t(Y; S'|s) + \mathcal{I}_t(Y; (A_1, \ldots, O_L)|S') \qquad \because S' = s \cup (A_0, O_1) \tag{35}$$

$$= \mathcal{I}_t(Y; S'|s) + \sum \mathbb{P}_t(s'|s; \pi)\mathcal{I}_t(Y; \tau^\pi(s')|s') \tag{36}$$

$$= \mathcal{I}_t(Y; S'|s) + \sum \mathbb{P}_t(s'|s; \pi)\mathscr{I}_t^\pi(s') \qquad \because \text{By definition of } \mathscr{I}_t^\pi(s') \tag{37}$$

$$= \mathcal{I}_t(Y; S'|s) + \mathbb{E}_\pi[\mathscr{I}_t^\pi(S')|s]. \tag{38}$$

$$\square$$

**Remark 1.** *Note that as a 'state' is the history of everything that has occured so far, this makes the decision process resemble an (acyclic) tree, where each 'state' can only be reached through a unique path, and can't be revisited again. This is good, as it is aligned with the problems we aim to resolve. But this may be not so good, because it probably makes learning $\mathscr{I}(\cdot)$ much harder.*

**Remark 2.** *During test-time inference, we need a value function that estimates $\mathscr{I}_t^\pi(h)$ so as to perform n-step information backup. But how should we learn $\mathscr{I}_t^\pi(h)$? Here it is important to distinguish the implicit conditioning on (the entire past training data) through the subscript $t$, and the explicit conditioning (i.e., in-context) on the observations made in the current interaction. Therefore, this will require an 'intermediate' phase where we will need pairs of $(s, \hat{\mathcal{I}}_t^\pi(Y; \tau^\pi(s)|s))$ from a held-out dataset to estimate $\mathscr{I}_t^\pi(s)$, which can then be used during the test time to do (n-step) backup. The important thing to note here is that the data from the 'intermediate' phase should not be used, ideally, to improve the language model again, as that would change the meaning of the implicit-conditioning on $t$. For example, if $\mathcal{I}_t^\pi(Y; \tau^\pi(s)|s)$ had a non-zero value and then we use $s$ to update the language model again, and if we assume everything to be perfect, then $\mathcal{I}_{t+1}^\pi(Y; \tau^\pi(s)|s)$ should be 0 as there is no new information to be gained from $s$.*

Due to the above challenges in estimating long-term information gain $\mathscr{I}_t^\pi(s)$, we resort to myopic version, where only one-step information gain backup is used, and $\mathscr{I}_t^\pi(s)$ is set to 0. This is akin to how even UCT does not take into account uncertainty of $Q$, it still does one-step/bandit uncertainty.

## C  Empirical Details

In Algorithm 1 we present the pseudo-code for IDTS. The steps highlighted in blue are the key differences from the typical tree-search/UCT.

For different domains, we defined $\text{Value}(s, \tau^\pi(s))$ differently. For coding domain, this corresponded to the the fraction of unit test passed by the latest solution available in $\tau^\pi(s)$. For LLF-bench, this comprised of the scalar feedback form the environment for how relevant is the latest recommendation to the movie that the user had in mind. For math domain, we set $\text{Value}(s, \tau^\pi(s)) = 0$, and treat the task as a pure exploration problem.

For all the domains, we set $n = 1$ for rollout in the $\text{Evaluate}$ function. Therefore, $\tau^\pi(s) = s'$. With this setting $\hat{\mathcal{I}}(Y; \tau^\pi(s) | \mathcal{T}, D)$, where in the algorithm $\mathcal{T}$ represents the current tree, can be equivalently expressed as $\hat{\mathcal{I}}_t(Y; s')$, where $t$ is the current iterate.

Results for the code, and math domain used `gpt-3.5-turbo-0125`, and for the LLF-bench `gpt-4-turbo-2024-04-09` was used.

---

**Algorithm 1:** The IDTS algorithm

1 **Function** `IDTSearch`($s_0$)**:**
2     Create tree $\mathcal{T}$ with root state $s_0$
3     **while** *within compute budget* **do**
4         $s \leftarrow \text{TreePolicy}(s_0)$
5         $v, i \leftarrow \text{Evaluate}(s)$
6         $\text{Backup}(s, v, i)$
7     **return** $\arg\max_{s \in \hat{\mathcal{T}}} V(s)$

8 **Function** `TreePolicy`($s$)**:**
9     **while** *$s$ is non-terminal* **do**
10         **if** *$s$ is not $K$-expanded* **then**
11             **return** $\text{Expand}(s)$
12         **else**
13             $s \leftarrow \text{BestChild}(s)$
14     **return** $s$

15 **Function** `BestChild`($s$)**:**
16     **return** $\arg\max_{s' \in \mathcal{C}_t(s)} V(s') + \lambda I(s')$

17 **Function** `Evaluate`($s$)**:**
18     (n-step) Rollout $\tau^\pi(s)$
19     $v \leftarrow \text{Value}(s, \tau^\pi(s))$
20     $i \leftarrow \mathcal{I}(Y; \tau^\pi(s) | \mathcal{T}, D)$
21     **return** $v, i$

22 **Function** `Expand`($s$)**:**
23     $A \leftarrow \text{LLM}(s, \mathcal{T}_t)$
24     $O \leftarrow \text{GetFeedback}(s, A)$
25     $s' = (s, A, O)$
26     $\mathcal{T} \leftarrow \mathcal{T} + \text{Child } s' \text{ added to } s$
27     **return** $s'$

28 **Function** `Backup`($s, v, i$)**:**
29     **while** *$s$ is not null* **do**
30         $\alpha \leftarrow N(s)/(N(s) + 1)$
31         $V(s) \leftarrow \alpha V(s) + (1 - \alpha)v$
32         $I(s) \leftarrow \alpha I(s) + (1 - \alpha)i$
33         $N(s) \leftarrow N(s) + 1$
34         $s \leftarrow \text{parent of } s$
35         $v \leftarrow r(s) + v$

---

**Practical Approximations:** Recall from 5 that we need to estimate $p(y|s', \mathcal{T}_t, D)$. This corresponds to epistemic uncertainty for the LLM. Several methods exists for estimating this [Lakshminarayanan et al., 2017, Osband et al., 2021], we make use of the ensemble approach where

$$p(y|s', \mathcal{T}_t, D) = \int p(y|s', \mathcal{T}_t, \theta) p(\theta|D) \mathrm{d}\theta \approx \frac{1}{M} \sum_{i=1}^{M} p(y|s', \mathcal{T}_t, \theta_i). \tag{39}$$

For extremely large models, ensembles can be created using multiple-low rank adapter instead [Malinin and Gales, 2020, Kuhn et al., 2023]. For our experiments, we simply use $M = 1$, such that $p(y|s', \mathcal{T}_t, D) \approx p(y|s', \mathcal{T}_t, \theta)$. Similarly for $p(y|\mathcal{T}_t, D) \approx p(y|\mathcal{T}_t, \theta)$. Computing an estimate information gain $\mathcal{I}_t(Y; s')$ now simply corresponds to estimating the difference in entropy of $p(y|s, \mathcal{T}_t, \theta)$ and $p(y|\mathcal{T}_t, \theta)$.

Further, conditioning on the information of the entire tree $\mathcal{T}_t$ might require very large context window for the LLMs. To avoid such long context, instead of conditioning on the content of all the nodes in $\mathcal{T}_t$, we condition only on the ancestor nodes of $s'$. Finally, we also note that that the estimator in (11) can result in negative values because of the sampling error. To avoid this, we clip the minimum value of the information gain to be 0.

