# OpenReview forum: "Information Directed Tree Search: Reasoning and Planning with Language Agents"
_NeurIPS.cc/2024/Workshop/BDU — NeurIPS BDU Workshop 2024 Poster_

### Official Review · Reviewer_PsGt · 2024-09-26
**Review of an IDTS for enhancing LLMs reasoning and planning capabilities**

**Rating:** 5
**Confidence:** 4

**Review:**

This paper presents an Information Directed Tree Search (IDTS) algorithm designed to enhance reasoning and planning capabilities of Large Language Models (LLMs) in complex tasks requiring multi-step reasoning and feedback integration.

By incorporating a Bayesian approach, the authors seek to guide the search process toward feedback that offers higher information gain, leveraging in-context learning to approximate this gain without explicit posterior updates. While IDTS shows some advantages over simple uniform search methods, it performs comparably to Monte Carlo Tree Search (MCTS), albeit exploring different paths.
The paper is well-structured and provides a solid theoretical foundation for the proposed method. The authors clearly define the problem, formalize their approach using Bayesian principles, and explain how in-context learning can be utilized to estimate information gain without explicit posterior updates. The inclusion of mathematical formulations and theoretical proofs enhances the clarity of the methodology.
The practical significance of the method is not fully established due to the limited empirical gains observed. The proposed IDTS performs comparably to MCTS but does not significantly outperform it. This raises questions about the added complexity versus the benefits gained.

Pros:
- Introducing information gain into tree search for LLMs is a creative idea.
- The paper provides rigorous theoretical backing, including proofs and formulations using information theory and Bayesian inference.
- The authors openly discuss the limitations of their approach and identify areas for future research.

Cons:
- IDTS does not show significant performance gains over MCTS in the experiments conducted.
- The paper lacks in-depth comparisons with other exploration strategies or uncertainty-guided algorithms.
- A deeper investigation into the estimation errors of information gain and alternative estimation methods would strengthen the contribution.
- The paper lacks comparisons with other advanced planning or search techniques beyond MCTS and simple uniform search methods.
    + Andy Zhou, Kai Yan, Michal Shlapentokh-Rothman, Haohan Wang, and Yu-Xiong Wang. 326 Language agent tree search unifies reasoning acting and planning in language models. arXiv preprint arXiv:2310.04406, 2023.
    + Shunyu Yao, Jeffrey Zhao, Dian Yu, Nan Du, Izhak Shafran, Karthik Narasimhan, and Yuan Cao. ReAct: Synergizing reasoning and acting in language models. In ICLR, 2023b.
    + Jason Wei, Xuezhi Wang, Dale Schuurmans, Maarten Bosma, Ed Chi, Quoc Le, and Denny Zhou. Chain of thought prompting elicits reasoning in large language models. arXiv:2201.11903, 2022.

The paper introduces a promising idea with a solid theoretical basis but falls short in empirical validation. It would benefit from providing more comprehensive experimental results and testing on additional benchmarks.

---

### Official Review · Reviewer_2tzU · 2024-10-07

**Rating:** 6
**Confidence:** 3

**Review:**

To enhance the quality of feedback sources for autonomous agents, this paper proposes a new algorithm called Information-Directed Tree Search (IDTS), aimed at improving the reasoning and planning capabilities of language models in complex tasks. Since these tasks often involve partially correct feedback, IDTS utilizes a Bayesian approach to prioritize the exploration of paths that can provide higher information gain, thereby increasing search efficiency. It performs well in tasks that require multi-step reasoning.

Weaknesses:
1. Although IDTS has shown some promise in encoding and LLF-Bench tasks, the performance improvement compared to MCTS is not significant. And it fails to clearly explain why the two perform similarly or differ in specific situations. The article should also provide a detailed explanation of the advantages and disadvantages of IDTS compared to other improved methods, such as search methods based on uncertainty estimation or reinforcement learning algorithms.

2. The feedback sources differ across tasks (programming, mathematics, natural language), but the article does not delve into how to dynamically adjust IDTS's search strategy based on the quality or type of feedback.

3. Contextual learning can update the posterior distribution and estimate information gain, but the specific working mechanism of contextual learning, especially its application in multi-step reasoning tasks, has not been explained in detail.

Overall, the method proposed in the article is quite novel, but there are still some details that need further clarification.

---

### Official Review · Reviewer_Kavn · 2024-10-08
**Review for Paper 118**

**Rating:** 8
**Confidence:** 4

**Review:**

This paper develops an Information Directed Tree Search (IDTS) method that leverages feedback on LLM responses to solve reasoning tasks. This is a common and practical application scenario for LLMs. The action is selected based on Information Gain (IG) for tree search. In the process, Bayesian inference and in-context learning are used together for estimation, thus overcoming the high-dimensional space issue that traditional RL frameworks cannot handle. The experiments show that the proposed method performs well on coding and movie recommendation tasks but falls behind the baseline on math tasks.

A few issues that I would be glad to see resolved:
Some notations and formulas are overly complex and not readable enough to me.
Did you try using feedback from other sources on the math task? This could help determine whether the relatively poor performance is related to the feedback or its nature.

Given that it is a workshop paper with a 4-page limit, I think the authors have conveyed their thoughts and approach well, so I suggest an acceptance.

---

### Official Review · Reviewer_SHFn · 2024-10-08
**Novel Information-Directed Tree Search for LLM Planning Shows Promise but Needs Further Analysis**

**Rating:** 7
**Confidence:** 3

**Review:**

**Summary**

Information Directed Tree Search (IDTS) is proposed as a new approach for inference-time planning on pretrained LLM responses. As a Bayesian approach, IDTS quantifies the information associated with different feedbacks and prioritizes the tree search towards feedback that provides higher information gain. For information gain estimation, in-context learning is combined with Bayesian inference. The effectiveness of the proposed method is evaluated in three different LLM reasoning tasks.

**Strengths:**
1. The idea of using information-directed sampling within a tree search algorithm for LLM-based problem solving is novel, to my knowledge.
2. The approach is explained in a clear and detailed way, both verbally and theoretically.
3. Experiments are conducted on mainstream LLM tasks, which are meaningful, and baselines are well-selected.
4. Results are promising, especially in tasks with corrective feedback.

**Weaknesses:**
1. The computational complexity of the proposed method could be compared with MCTS and discussed in the paper more clearly and in more detail.
2. The performance gain over MCTS is limited, but it is promising. The results could be analyzed in more depth. Why do IDTS and MCTS perform similarly? Under what conditions might IDTS outperform baselines?

---

### Decision · Program_Chairs · 2024-10-09

Accept (Poster)